# Bio-Drying of Municipal Wastewater Sludge: Effects of High Temperature, Low Moisture Content and Volatile Compounds on the Microbial Community

Vladimir Mironov [1,*], Ivan Moldon [1], Anna Shchelushkina [1], Vitaly Zhukov [1] and Nataliya Zagustina [2]

1 Winogradsky Institute of Microbiology, Research Center of Biotechnology, Russian Academy of Sciences, 119071 Moscow, Russia
2 Bach Institute of Biochemistry, Research Center of Biotechnology, Russian Academy of Sciences, 119071 Moscow, Russia
* Correspondence: 7390530@gmail.com; Tel.: +7-(903)-7390530

**Abstract:** This study examined microbiological processes during the bio-drying of municipal wastewater sludge (WS) from the waste treatment facilities of the Moscow region (Russia). In just 21 days of bio-drying, the moisture content of the mixture of WS and wood chips decreased by 19.7%. It was found that members of the genus *Bacillus* were the main organic matter destructors. In the period from 7 to 14 days, the rates of organic matter mineralization and moisture loss were the highest, and bacteria of the genus *Bacillus* dominated, accounting for 43.5 to 84.6% of the bacterial community with a total number of $1.20 \, (\pm 0.09) \times 10^6$ to $6.70 \, (\pm 0.44) \times 10^5$ gene copies $\mu g^{-1}$. The maximum number of *Amaricoccus* was (15.7% of the total bacterial community) in the middle of bio-drying. There was an active accumulation of nitrate nitrogen due to the oxidation of nitrogen-containing substances during the same period of time. Bacteria of the genera *Sphingobacterium*, *Brevundimonas*, *Brucella*, *Achromobacter* and fungi of the genus *Fusarium* dominated in the biofilter, which removed volatile compounds from the waste air by 90%. The obtained results allow to model the further intensification of bio-drying, as well as its efficiency and safety.

**Keywords:** wastewater sludge; bio-drying; moisture loss; mineralization; biofiltration; community of microorganisms

## 1. Introduction

According to Business Guide [1], there are about 100 million tons of wastewater sludge (WS is a mixture of raw sludge and excess activated sludge) with a moisture content of 98% formed annually in Russia. The treatment and disposal of WS play role in environmental pollution, human health risks and the high cost of their disposal [2]. Due to the significant content of heavy metals and limitations of application in agriculture, thermal processes have become an attractive option for WS treatment in many countries [3]. However, due to the high water content of WS, a pre-drying step is required before incineration. Despite the improvement of mechanical dewatering methods, including centrifugation and the usage of vacuum filters or belt filter presses, the moisture content in WS after mechanical dewatering at most treatment plants still reaches 80–85% [4–7]. Dewatering plays a crucial role in WS management and accounts from 49 to 53% of total operational costs [5]. Recently, bio-drying has been brought to the forefront of pre-treatment of high-moisture organic waste [5,8].

Bio-drying is a technology that removes water from a material using biological heat generated by the aerobic microbial decomposition of organic matter [5]. The initial mixture of materials for bio-drying is characterized by a high diversity of microbiota. Thus, bio-drying is a highly dynamic process in which bacteria and micromycetes play an important role. Due to the biologically generated heat by the microbiota (bio-heat) and mechanical

operations, bio-drying passes through the mesophilic, thermophilic and cooling stages, which are followed by the succession of the microbial community. Self-heating of the material changes the microbial community. Therefore, this affects the decomposition of organic matter and the rate of water evaporation. In addition, microbial metabolism leads to the formation of water, which affects the efficiency of bio-drying. Maintaining the balance between the intensity of moisture removal and the decomposition of organic matter (OM) is an important feature of bio-drying.

Bio-drying is primarily the result of the metabolic activity of microorganisms. Thus, several works from the review article [5] studied both water removal and changes in bacterial communities during bio-drying. They reported about composition of bacterial [9–14], fungal [13–15] and archaeal [16] communities during composting, i.e., an analogical process based on aerobic biodegradation. Despite previous studies of microbial populations during the composting process, the change in bacterial communities that play key roles in the bio-drying process has not been fully studied.

Mironov et al. [12] studied the succession inside the microbial community during the joint composting of anaerobically treated WS (AnWS) and wood chips. This research showed the predominance of bacteria up to 82–98% during all stages of composting, and the feature of this community was the almost complete absence of *Bacillus* members. Similar results were obtained by Wang et al. and Galitskaya et al. [17,18]. At the same time, members of genus *Bacillus* are some of the most common microorganisms that colonize compostable material, as shown in both classical microbiological and molecular studies [5,11,13,14,19,20]. *Bacillus* spp. are often used as a microbial compost accelerators for various wastes [21–24]. Unfortunately, conflicting data on the predominance of the genus *Bacillus* do not properly assess their contribution to the degradation of OM and the prospects for bioaugmentation to accelerate bio-drying of WS.

Spores of *Bacillus* species have a significant resistance to high temperature combined with low moisture content [25]. Most of the factors that are potentially responsible for spore tolerance to high temperature and low moisture content as well as sporulation temperature are not properly understood. Today, the only known factor that promotes the resistance of spores to desiccation is the protection of DNA by acid-soluble spore proteins. The study on bio-drying by Cai et al. [5] showed *Bacillus* spp. dominated by a moisture content level of 54.7%. Therefore, *Bacillus* spp. are able to get key positions even at a similarly low moisture content level. Perhaps, this is connected with transition of *Bacillus* spp. into dormant forms due to unfavorable conditions.

Our assumption is that during bio-drying under high temperature conditions (thermophilic stage) and low moisture content (20–40% by weight), members of the genus *Bacillus* may have a significant advantage (dominance in the community).

The purpose of this research is to study (i) the effect of high temperature and low moisture content on the succession of the microbial community, (ii) the contribution of *Bacillus* members into degradation of organic matter, and (iii) the microbial community of the biofilter providing purification of waste air from bio-drying.

## 2. Materials and Methods

### 2.1. Substrate

Experimental bio-drying of municipal WS and lignocellulosic wastes with simulation of production conditions was carried out using dehydrated WS obtained from the wastewater treatment plants in the Moscow region (Russia). Before bio-drying, WS was mixed with wood chips (fraction 5–10 mm) of hardwoods in a WS:chips ratio of 2:1.35 by weight to achieve the desired porosity and moisture content [26]. The characteristics of the initial mixture are presented in Table 1.

**Table 1.** The principal physical, chemical, biological and physicochemical parameters of the initial mixture of WS and wood chips.

| Parameter | Units | Value * | Optimal Limits |
|---|---|---|---|
| pH | pH units | $6.28 \pm 0.50$ | 6.5–8.0 [27] |
| Moisture content | % | $53.10 \pm 2.12$ | 50–60% [26,27] |
| Air-filled porosity | % | $57.1 \pm 0.2$ | 30–60% [26] |
| Total Kjeldahl nitrogen (N) | % | $1.76 \pm 0.14$ | |
| $NH_4^+$ | mg kg$^{-1}$ | $1618.21 \pm 129.46$ | |
| $NO_3^-$ | mg kg$^{-1}$ | $319.08 \pm 25.5$ | |
| Organic matter (OM) | % | $74.14 \pm 1.48$ | |
| Total organic carbon (C) | % | $37.07 \pm 1.48$ | |
| C/N | | $21.33 \pm 1.62$ | 20–30 [26–28] |
| Germination index (GI) | % | $59.56 \pm 4.76$ | |
| Pb | | $27.30 \pm 1.94$ | 130 [29] |
| As | | $0.0 \pm 0.0$ | 10 [29] |
| Cd | | $0.0 \pm 0.0$ | 2 [29] |
| Ni | | $5.85 \pm 0.46$ | 80 [29] |
| $Cr^{3+}$ | mg kg$^{-1}$ | $11.31 \pm 0.85$ | 6 [29] |
| Mn | | $85.29 \pm 5.37$ | 1500 [29] |
| Zn | | $509.65 \pm 25.99$ | 220 [29] |
| Cu | | $37.44 \pm 1.68$ | 132 [29] |

*—average value $\pm$ standard deviation of three replicates.

### 2.2. Experimental Setup

The bio-drying process was carried out for 21 days in a test (laboratory) setup simulating the conditions of aerobic solid phase biodegradation according to the scheme described in Figure 1 [13]. The bio-drying time of 21 days was chosen similarly to that reported by Cai et al. [5].

The working volume of the reaction chamber was 10 dm$^3$; six chambers were used simultaneously for two parallel experiments. The substrate was forcibly aerated with supplying 0.04 L of air min$^{-1}$ kg$^{-1}$ OM due to the vacuum created using the blower and the inflow of atmospheric air into the lower layer of the substrate in the chamber, which corresponded to the model of natural aeration of compost heaps [30]. The experiments were carried out under laboratory conditions at a temperature of $18.4 \pm 1.8\,°C$ and an air humidity of $26.4 \pm 2.2\%$.

Measurement of the gas composition of the waste air was carried out continuously using a gas analyzer MAG-6 S-1 (Eksis, Moscow, Russia): $O_2$ ($0$–$30 \pm 0.4$ vol.%), $CO_2$ ($0$–$10 \pm 0.1$ vol.%), $NH_3$ ($0$–$20 \pm 4$ mg m$^{-3}$), $H_2S$ ($0$–$10 \pm 2$ mg m$^{-3}$). The temperature of the composted substrate changed due to self-heating, regardless of the ambient temperature. The temperature in each chamber was maintained independently at the current substrate temperature $\pm 0.2\,°C$ using a heating element and an IVTM-7/2S temperature meter-regulator (Eksis, Russia). The temperature in the chamber was measured hourly automatically using sensors located in the substrate.

The biofilter for treatment of waste air consisted of a glass cylinder 10 cm high and 3 cm in diameter, loosely filled with an inert carrier, undrawn nylon fiber with an irregular surface structure. This nylon fiber formed a mechanical packing material with a specific surface area of 17–20 m$^2$ m$^{-3}$. The waste air from the WS bio-drying was supplied through an air flow regulator (rotameter) RMS-A-0.035 GUZ-2 (Pribor-M, Arzamas, Russia) into the biofilter from top to bottom, and the air retention time in the biofilter was 5–10 s. Cleared air after biofilter passed to a flask with 0.01 M potassium phosphate buffer of pH 7.0, providing bubbling of the solution. To irrigate the biofilm carrier, the solution from the flask was supplied using a peristaltic pump from above, dropwise, at a rate of 5–6 mL per minute. Irrigation solution was changed every 2 weeks.

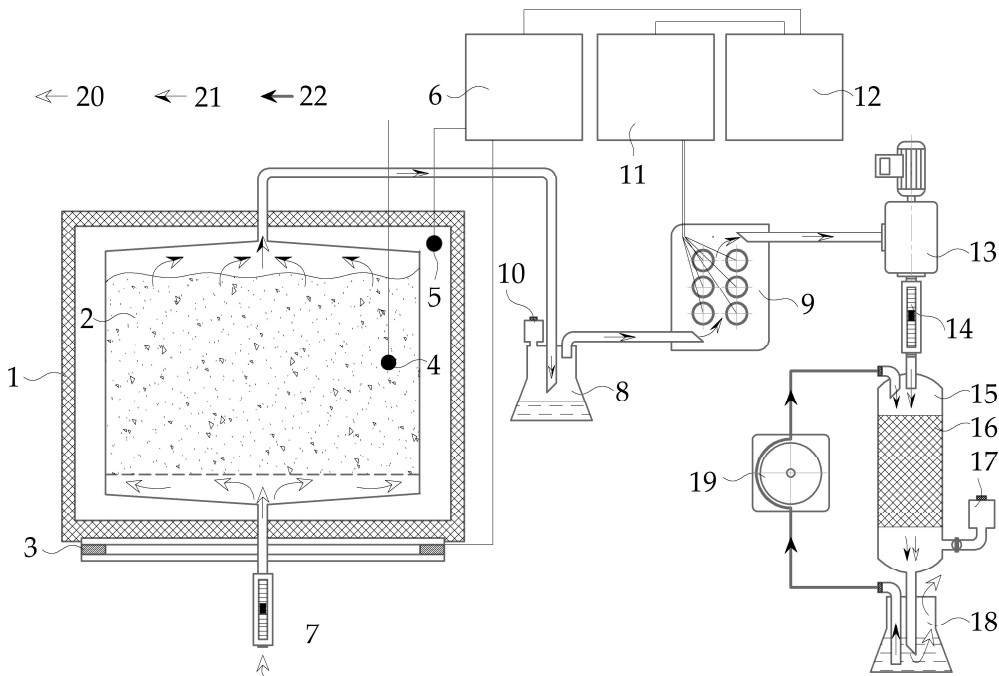

**Figure 1.** Schematic image of a test (laboratory) setup simulating the conditions of aerobic solid phase biodegradation: reaction chamber (1); compost mixture (2); heating element (3); temperature sensors (4, 5); temperature meter-regulator (6); rotameter (7, 14); flask (8, 18); gas sensors (9); sampling port (10, 17); gas analyzer (11); computer (12); blower (13); glass cylinder (15); carrier with biofilm (16); peristaltic pump (19); atmospheric air (20); waste air (21); and irrigation solution (22).

At the beginning of the work, the biofilter was inoculated with a suspension of microorganisms previously isolated from working biofilters. It included the following bacteria species: *Pseudomonas putida*, *P. oleovorans*, *P. fluorescens*, *Comamonas testosteroni*, *Rhodococcus* sp., *Pseudomonas* sp.

Chromatographic analysis to assess the content of volatile organic compounds (VOCs) was performed on a nonpolar capillary column OV-101 using a gas chromatograph (GS) Khromatek-5000 (Khromatek, Yoshkar-Ola, Russia) with a flame ionization detector (FID). The air was sampled using 1000 μL Hamilton type gas chromatography microsyringes. Waste air was sampled from an inlet sampling port after a reaction chamber. The cleared air after biofilter was sampled in an outlet sampling port. The combined error of the chromatograph detector and the software method, equal to ±10%, was taken as the measurement error due to low concentrations of VOCs in waste air and consequently low peak areas in GS chromatograms.

### 2.3. Physicochemical Studies

Samples for the study of physicochemical and biological parameters were taken, starting from the first day of the experiment and then on days 7, 14 and 21. Mechanical mixing of the substrate was immediately carried out each time after sampling.

The following parameters were measured in all samples according to generally accepted and modified methods: pH using an ANION 4150 laboratory analyzer (Ifraspak-Analit, Novosibirsk, Russia), moisture content using the thermogravimetric method on an EvLAS-2M moisture analyzer (Sibagropribor, Krasnoobsk, Russia), total organic carbon (C, %) by thermogravimetric method using a muffle furnace PM-16M-1200 (MT-Etalon, Moscow, Russia) [31], total nitrogen (protein and ammonium) according to Kjeldahl (N, %) in dry matter using automatic titrator Easy Plus (Mettler Toledo, Greifensee, Switzerland), water-soluble ammonium ion ($NH_4^+$) and nitrate ($NO_3^-$) nitrogen [32], GI [33], C/N ratio, air-filled porosity [26]. A 10 g sample resuspended in 300 mL distilled water was used to

determine water solubles, pH and GI [34]. The concentration of $NH_4^+$ and $NO_3^-$ nitrogen was determined using a spectrophotometer Hach Lange DR 5000 (Hach Lange, Dusseldorf, Germany): ammonium with Nessler's reagent and nitrate were measured by the method based on cadmium reduction. The compost effect on plant growth was evaluated using seed germination and root length of the radish (*Raphanus sativus*) based on the calculation of the GI.

### 2.4. Microbial Counting with Real-Time PCR (qPCR-RT)

To isolate DNA from the bio-drying material and biofilter, a set of "DNeasy PowerSoil Kit" (Qiagen, Hilden, Germany) was used. Real-time PCR was performed in PCR buffer-B (Syntol, Moscow, Russia) with the presence of the intercalating dye SYBR Green I and the passive reference dye ROX Syntol using the real-time PCR system CFX96 TouchTM (Bio-Rad, Hercules, CA, USA).

To quantify the content of archaea, bacteria and fungi in the studied samples, the following primer systems were used for the 16S rRNA gene and the internal transcribed spacer (ITS): Arch 967F/ Arch-1060R for archaea [35], Eub338F/Eub518R for bacteria [36], FR1/FF390 for fungi [37]. Quantification of each sample was carried out twice. To count the number of phylotypes in the analyzed samples, the signal received in the sample was compared with a standard curve. To construct standard curves, sequential dilution series of a standard sample were used. The target PCR fragment was used as a standard sample and was previously purified using the WizardSV Gel and PCR Clean-Up System (Promega, Madison, WI, USA) and cloned in pGEM-T vector Promega. The number of target phylotypes in the analyzed samples was calculated per 1 µg of substrate or 1 µg of raw sediment from the biofilter and expressed in gene copies µg$^{-1}$.

### 2.5. Profiling of Prokaryotic and Fungal Communities Based on 16S rRNA Gene and Internal Transcribed Spacer (ITS)

To isolate the total DNA, the "PowerSoil DNA Isolation Kit" (MoBio, Carlsbad, CA, USA) was used. The procedure was carried out according to the protocol recommended by the manufacturer. DNA was stored at $-20\,°C$. The purified DNA sample was used as matrix for PCR with universal primers for V3–V4 sections of the prokaryotic 16S rRNA gene: 319F (5′-ACTCCTACGGGAGGCAGCAG-3′) and 806R (5′-GGACTACHVGGGTWTCTAAT-3), as well as primers for ITS fungi taxa: 1737F: (5′-GGAAGTAAAAGTCGTAACAAGG-3′) and 2043R: (5′-GCTGCGTTCTTCATCGATGC-3′). The primers were supplemented with identifying oligonucleotide sequences for the MiSeq (Illumina, San Diego, CA, USA) sequencer. Sample preparation and sequencing were carried out according to the manufacturer's recommendations. UPARSE [38] was used to control the quality of the reeds. Then, the reeds were grouped in operational taxonomic units (OTUs) according to the level of similarity of 97% using USEARCH [39]. The taxonomic position of the sequence for each OTU was determined using the "RDP classifier" [40] and "Blast" [41].

### 2.6. Statistical Data Analysis

Throughout the study, three replicates of each measurement were performed. The data were subjected to analysis of variance (ANOVA) using the least significant difference test. The test of significance was determined at $p < 0.05$. The results are presented in the form of an "average value ± standard deviation" of three replicates [42].

### 3. Results

#### 3.1. Dynamics of Physicochemical Parameters

According to the dynamics of temperature and respiration in the first 7 days of bio-drying, there were two transitional periods (Figure 2).

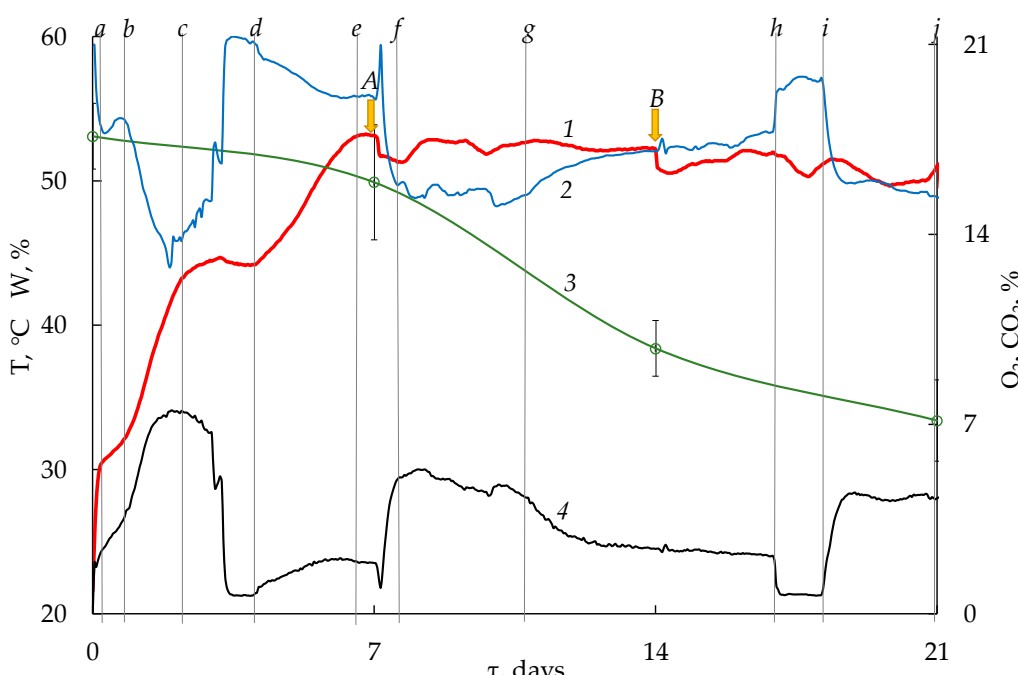

**Figure 2.** Temperature, moisture content and respiratory profiles during bio-drying of WS with wood chips (built on average hourly values with a standard deviation of values less than 10% at $p < 0.05$): 1—substrate temperature; 2, 4—the content of oxygen and carbon dioxide in the waste air, respectively; 3—moisture content in the substrate; A, B—mixing of the substrate; a–j—boundaries of transition periods (a–d—mesophilic stage; d–j—thermophilic stage).

After a rapid increase in temperature at the first 4 h from 21 to 30 °C (line *a*), followed by the consumption of $O_2$ and formation of $CO_2$ (18.2 and 2.16 vol.%, respectively), then the slowing down of biochemical reactions and decreases in the absorption of $O_2$ and in heat release were observed. At the same time, the temperature was at the level of 30–32 °C over the next 17 h, and the concentration of $O_2$ and $CO_2$ was 18 and 3 vol.%, respectively. Starting from 22 h (line *b*) till 52 h of bio-drying, the process was intensive; the temperature increased to 43 °C; the $O_2$ concentration decreased to 14 vol.%; and $CO_2$ increased to 7 vol.%. The second distinct slowdown of reactions began at 52 h (line *c*) of bio-drying and continued for 45 h (line *d*). At this time, the temperature of the substrate was near 43 °C due to thermal inertia and low thermal conductivity of the substrate. At this time, $O_2$ absorption and $CO_2$ production were very low. Furthermore, at 98 h of bio-drying, the beginning of a second increase in temperature, $O_2$ absorption and $CO_2$ formation were observed, and the temperature rose to a maximum of 53 °C by 156 h (line *e*). At the same time, the second stage of heat release was not accompanied with a significant absorption of $O_2$ and formation of $CO_2$. Their concentrations practically did not change and were 19 and 2 vol.%, respectively.

From day 7 to 21 of bio-drying, the temperature remained within 50–52 °C, but respiration and moisture loss in different periods were significantly different too. Thus, after mixing the substrate (point *A* on Figure 2) in the period from day 7 to 11 (section *f*–*g*), a significant absorption of $O_2$ (concentration of about 15–16 vol.%) and release of $CO_2$ (4–5 vol.%) were observed. During this period, the largest decrease in moisture mass in the substrate by 11.5% and a significant formation of ammonia were detected. The average daily concentrations of ammonia ranged from 13.92 ± 1.67 to 33.36 ± 2.84 mg m$^{-3}$ (Figure 3a) with the maximum hourly values of 320 mg m$^{-3}$.

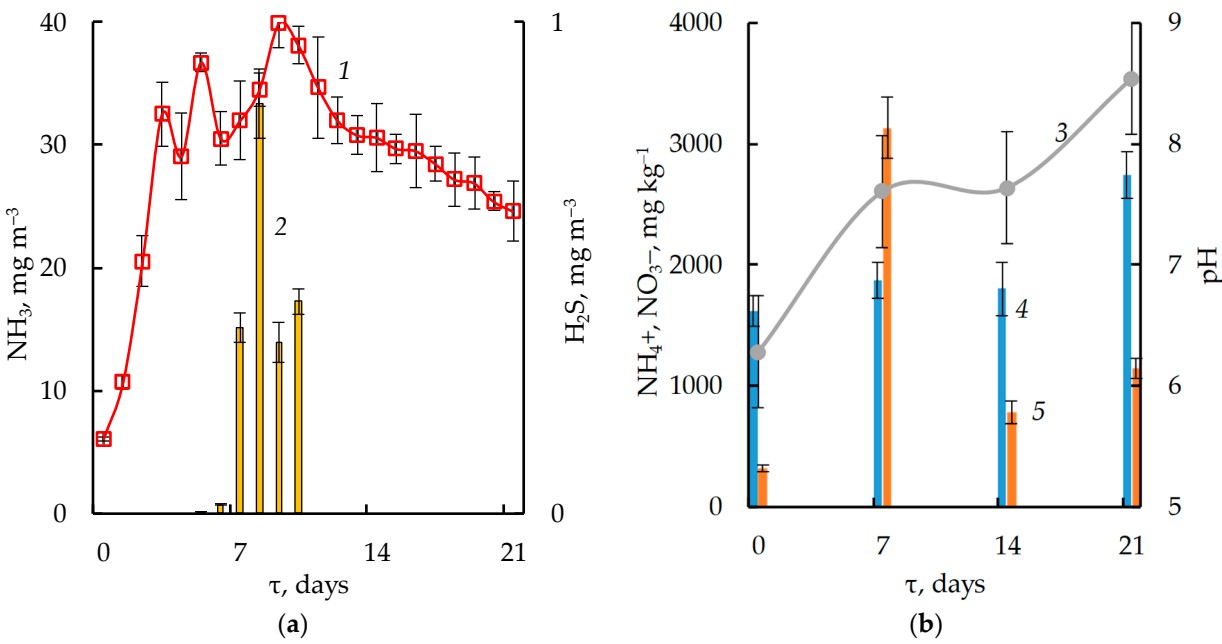

**Figure 3.** Changes in chemical and physical parameters and stability indicators: (**a**) the gas composition of the waste air: 1—$H_2S$, 2—$NH_3$; (**b**) 3—pH, content; 4—$NH_4^+$; and 5—$NO_3^-$ (error bars are the mean ± standard deviation of three replicates).

Further, the intensity of the reactions gradually decreased. Thus, during this period the average $O_2$ and $CO_2$ values were 17–19 vol.% and 2–3 vol.%, respectively. From the 18th day for 28 h (section *h–i*), almost complete cessation of biodegradation was observed. The concentration of $O_2$ increased to 20.5 vol.%, and $CO_2$ decreased to 0.5%. Later on, from day 18 to 21 (section *i–j*), the same rapid increase in $O_2$ absorption (decreased to 15 vol.%) and $CO_2$ formation was observed. Such dynamics of biodegradation process at practically constant temperature is probably based on the consistent succession of the microbial community, whose members are more dependent on changes in the composition of specific substrates than on temperature.

Concentration of $H_2S$ rapidly increased in the first 3 days to $0.81 \pm 0.07$ mg m$^{-3}$ and reached its maximum value of $1.00 \pm 0.05$ mg m$^{-3}$ on day 9 (Figure 3a).

The concentration of $CO_2$ in the substrate averaged 2.80 vol.% with a maximum of 7.51 vol.%. The absorption of oxygen was directly proportional to the formation of $CO_2$ and increased during the rapid temperature rising of the substrate. Absorption of oxygen was 5–6 h ahead of temperature growth, which might be connected with thermal inertness and low thermal conductivity of the substrate. At the same time, there were periods when significant respiration was not associated with temperature changes (sections *f–g* and *i–j* on Figure 2).

The bio-drying of WS increased the content of nitrate from $319.08 \pm 25.53$ to $1144.59 \pm 80.12$ mg kg$^{-1}$ and ammonium ion from $1618.21 \pm 129.46$ to $2739.74 \pm 191.78$ mg kg$^{-1}$ (Figure 3b) which is due to the processes of nitrification and ammonification, respectively.

Bio-drying increased the pH of the substrate from $6.28 \pm 0.50$ to $8.54 \pm 0.60$ due to the active ammonia formation at the end of the process (Figure 3b).

Based on the results of processing for 21 days, a substrate was obtained with the following main characteristics (Table 2).

**Table 2.** The principal physical, chemical, biological and physicochemical parameters of the substrate on day 21.

| Parameter | Units | Value * |
|---|---|---|
| pH | pH units | 8.54 ± 0.60 |
| Moisture content | % | 33.40 ± 3.20 |
| Air-filled porosity | % | 70.8 ± 0.5 |
| Total Kjeldahl nitrogen (N) | % | 2.00 ± 0.14 |
| $NH_4^+$ | mg kg$^{-1}$ | 2739.74 ± 191.78 |
| $NO_3^-$ | mg kg$^{-1}$ | 1144.59 ± 80.12 |
| Organic matter (OM) | % | 57.4 ± 1.15 |
| Total organic carbon (C) | % | 28.70 ± 1.15 |
| C/N | | 14.35 ± 2.44 |
| Germination index (GI) | % | 70.60 ± 4.94 |
| Pb | | 31.28 ± 1.81 |
| As | | 0.0 ± 0.0 |
| Cd | | 0.0 ± 0.0 |
| Ni | | 6.26 ± 0.36 |
| $Cr^{3+}$ | mg kg$^{-1}$ | 12.10 ± 0.39 |
| Mn | | 92.80 ± 6.77 |
| Zn | | 487.99 ± 39.53 |
| Cu | | 40.98 ± 1.43 |

*—average value ± standard deviation of three replicates.

*3.2. Molecular Genetic Research on the Taxonomic Diversity of Microbiota of the Substrate during Bio-Drying*

3.2.1. General Parameters of the Microbial Community

The largest number of microorganisms was in the initial mixture (Figure 4): bacteria had 1.0 (±0.06) × 10$^7$ gene copies μg$^{-1}$; fungi had 2.70 (±0.02) × 10$^6$ gene copies μg$^{-1}$. The total number of microorganisms decreased on day 7 (1.21 (±0.09) × 10$^6$ gene copies μg$^{-1}$) and had been decreasing until day 14 (6.81 (±0.44) × 10$^5$ gene copies μg$^{-1}$), though it grew slightly on day 21 (7.52 (±0.07) × 10$^5$ gene copies μg$^{-1}$). On these days, number of bacteria was 1.20 (±0.09) × 10$^6$ gene copies μg$^{-1}$ on day 7; 6.70 (±0.41) × 10$^5$ gene copies μg$^{-1}$ on day 14; and 6.30 (±0.04) × 10$^5$ gene copies μg$^{-1}$ on day 21. Thus, bacterial counts always reduced during bio-drying. At the same time, number of fungi was 1.60 (±0.52) × 10$^2$ gene copies μg$^{-1}$ on day 7; 1.00 (±0.01) × 10$^4$ gene copies μg$^{-1}$ on day 14; and 1.2 (±0.03) × 10$^5$ gene copies μg$^{-1}$ on day 21. Therefore, the minimal fungal counts were on day 7, and then, it increased by an order of magnitude every week. During the entire process, the presence of archaea changed little: in the initial biomass, their number was 8.80 (±0.41) × 10$^4$ gene copies μg$^{-1}$; and 2.00 (±0.07) × 10$^3$ gene copies μg$^{-1}$ on day 21. However, archaeal counts were always two orders of magnitude less than the bacterial ones.

3.2.2. Fungal Community of WS during Bio-Drying

The fungal community was quite variable at all stages of the process due to frequently changing conditions. Members of the divisions *Ascomycota*, *Mucoromycota* and *Basidiomycota*, which amounted up to 99% of the entire fungal community, played a key role in bio-drying. In the initial mixture, genera of the *Ascomycota* dominated: *Humicola* 50.1%, *Enterocarpus* 15.2%, *Pseudogymnoascus* 11.6%, *Leuconeurospora* 1.4%, and there was one from the *Mucoromycota*: *Mortierella* 5.8% (Figure 5). On day 7, the fungal counts were less than 1.60 (± 0.52) × 10$^2$ gene copies μg$^{-1}$, and *Pseudogymnoascus* accounted for 99.5%. Then, on day 14, *Pseudogymnoascus* was less than 0.3%, and a new *Mucoromycota* genus *Mucor* dominated (48.9%). At the same time, *Debaryomyces* was the second largest in the community (37.2%); *Melanocarpus* (*Ascomycota*) was 5.5%; and the first basidiomycete *Guehomyces* (4.7%) appeared. On day 21, *Pseudogymnoascus* dominated again (53.6%); *Orbicula* (*Ascomycota*) was the second largest (25.9%); *Mucor* was the third (12.8%). Ascomycetes *Gymnascella* and *Penicillium* accounted for 4.4 and 3.0%, respectively.

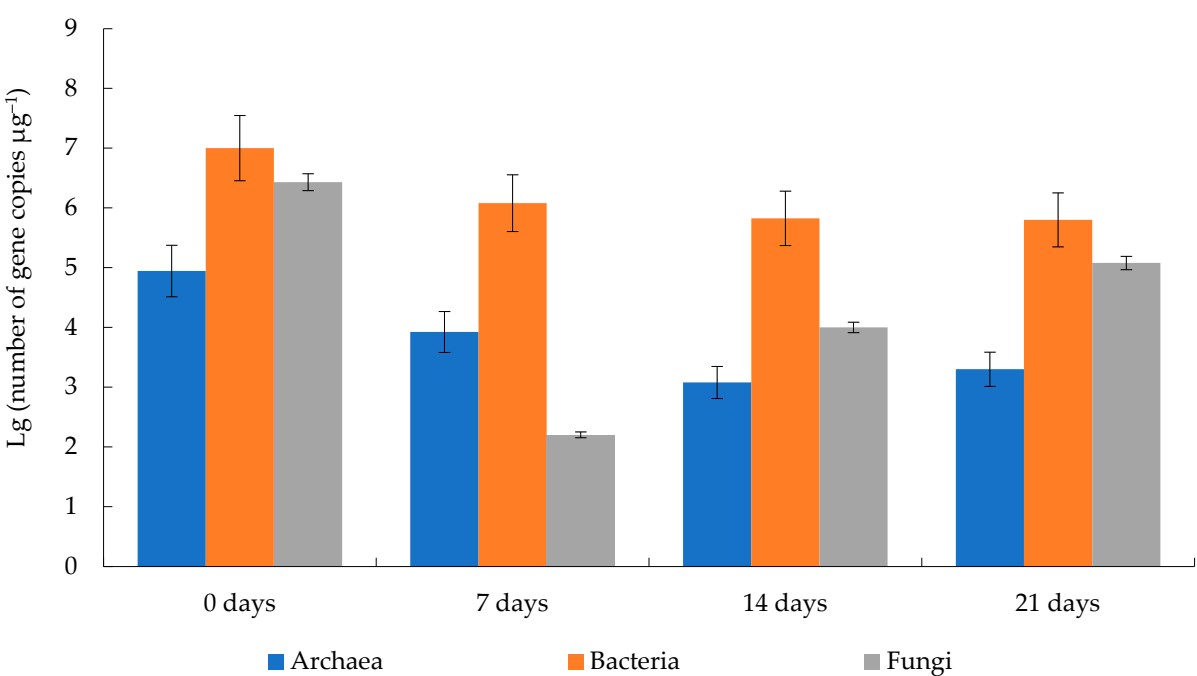

**Figure 4.** Change in the number of microorganisms, calculated from the number of copies of the 16S rRNA gene of bacteria and archaea and ITS of fungi (error bars are the mean ± standard deviation of three replicates).

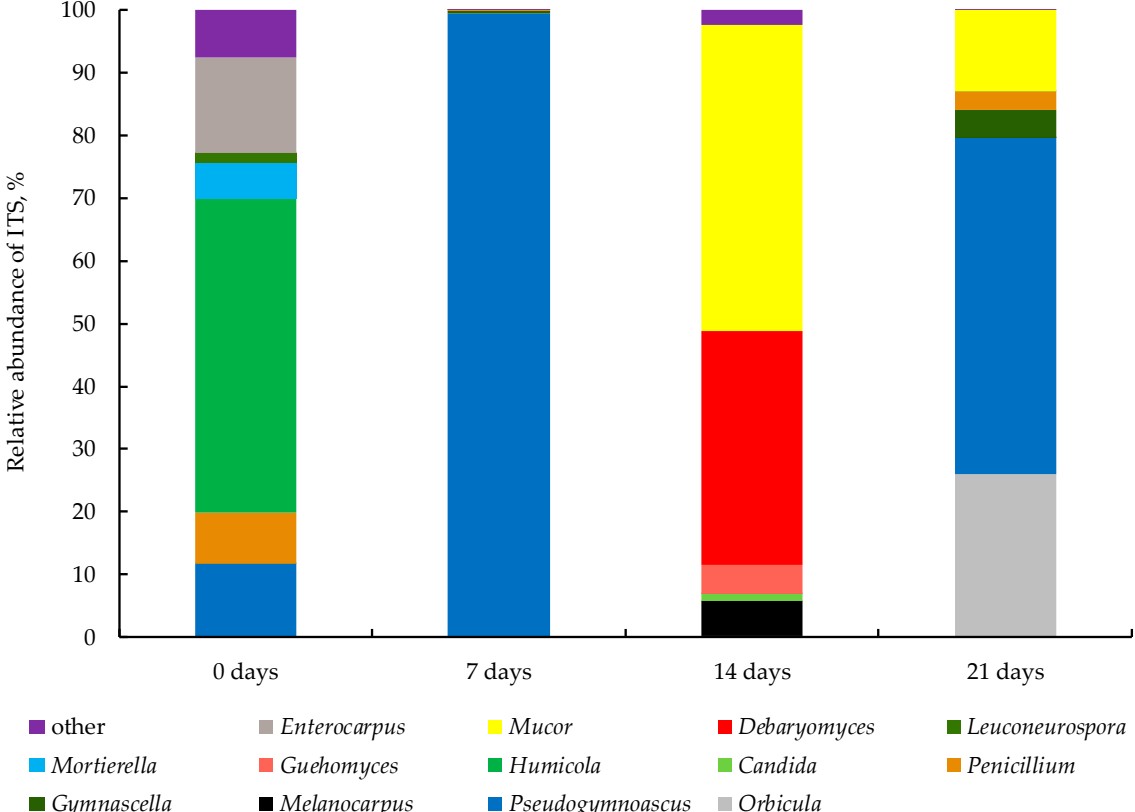

**Figure 5.** Genus-level composition of the fungal community (genera or higher taxa of fungi that constituted more than 1% of the fungal community are represented; the rest are grouped under the name "other").

### 3.2.3. Bacterial Community of WS during Bio-Drying

In the initial mixture, the dominant genera of bacteria were *Prevotella* 28.7%, *Simplicispira* 21.5%, *Shuttleworthia* 9.1% and *Pseudomonas* 7.2%, and minor genera were *Stenotrophomonas*, *Eoetvoesia*, *Lysobacter*, *Neochlamydia* and *Enterococcus*; the amount of each was less than 2% (Figure 6). At the thermophilic stage on days 7, 14 and 21, the dominant genus of bacteria was *Bacillus*: 43.5, 84.6 and 86.6% of the total bacterial community, respectively. The *Amaricoccus* increased from 15.7 on day 7 and decreased to 2.2% on day 14. Members of the genera *Clostridium* and *Romboutsia* on day 14 accounted for 1.3 and 1.7%, respectively.

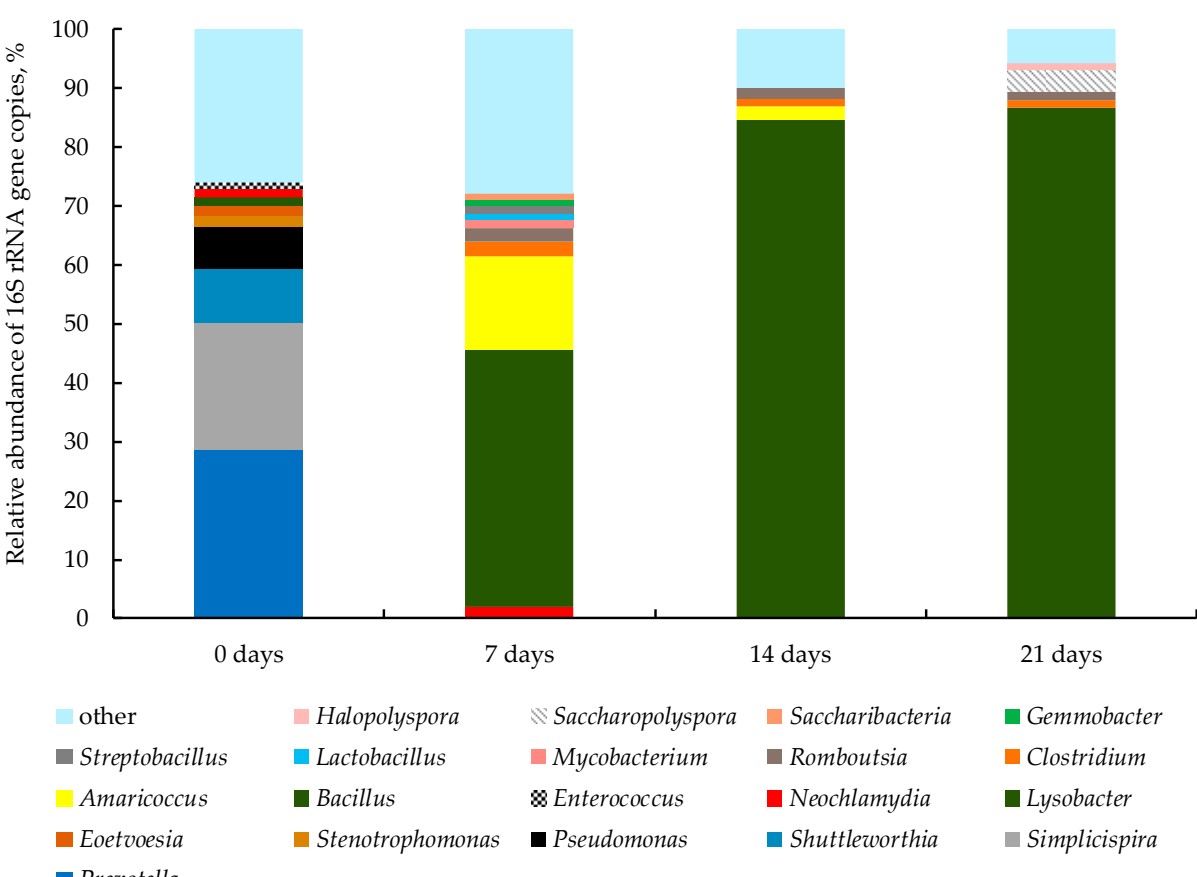

**Figure 6.** Genus-level composition of the prokaryotic community (genera or higher prokaryotic taxa that constituted more than 1% of the prokaryotic community are represented; the rest are grouped under the name "other").

### 3.2.4. Taxonomic Diversity of Biofilter Microbiota

Community profiling for fungal ITS and prokaryotic 16S rRNA gene fragments revealed more than 78 genera of bacteria and 3 genera of micromycetes. As a result, the total microbial abundance of the biofilter was 2.76 ($\pm$0.01) $\times 10^{10}$ gene copies $\mu g^{-1}$, among which 87.6% were bacteria, 12.3% fungi and 0.7% archaea.

The most abundant genera were the following: *Sphingobacterium* 44.5%, *Brevundimonas* 22.5%, *Brucella* 7.9%, *Achromobacter* 4.9%, *Pseudomonas* 2.3%, *Bacillus* 2.0%, *Clostridium* 1.5%, *Acinetobacter* 1.5% (Figure 7).

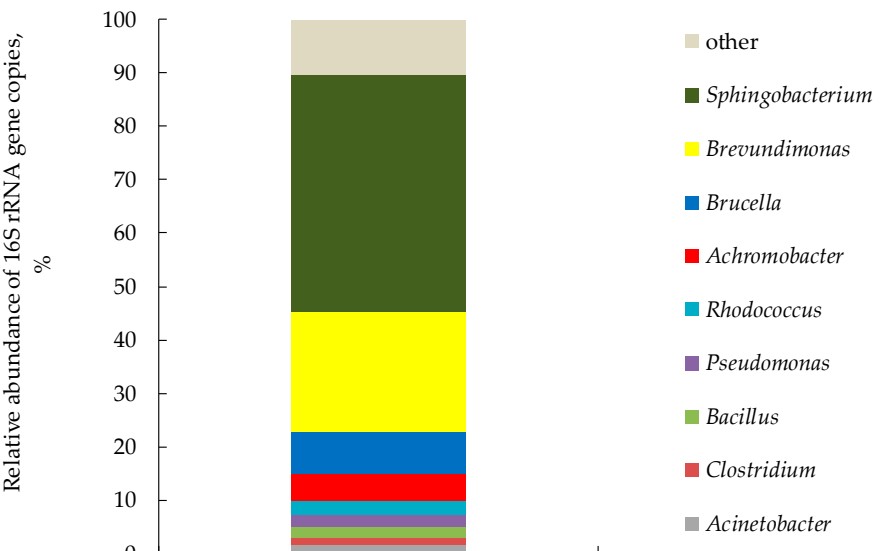

**Figure 7.** Genus-level composition of the prokaryotic community of the biofilter (genera or higher prokaryotic taxa that constituted more than 1% of the prokaryotic community are represented; the rest are grouped under the name "other").

## 4. Discussion

The microbial community in the initial mixture primarily used easily available substrates, and the temperature increased to 54 °C as a result of their vital activity. The greatest abundance of the microbial community was observed in the initial mixture. There were 1.28 ($\pm$0.06) $\times 10^7$ gene copies μg$^{-1}$ which is due to the presence of mesophiles before the temperature reached extreme values. Starting from day 7, the number of all domains has dramatically decreased by an order of magnitude. Perhaps, this is connected with the change of dominant communities, as well as with exhaustion of simple organic substances, gradual decrease in moisture and stable increase in temperature up to 50–54 °C. In the period from 7 to 14 days, the number of microorganisms continued to decrease. However, it grew slightly by day 21. Most likely, this indicates a gradual accumulation of temperature-adapted organisms that are able to use hardly decomposable substrates. The formation of ammonia and hydrogen sulfide in the first 7–10 days was probably associated with the intensive decomposition of proteins from dead microorganisms. Similar dynamics of ammonia emission, as well as a process of ammonification and nitrification, were also observed during the composting of AnWS [12]. The constant presence of ammonium ion in the substrate is a consequence of the decomposition of nitrogen-containing compounds by heterotrophic microaerophilic and anaerobic members of the putrefactive microbiota of *Bacillus* and *Clostridum*. The further increase in ammonium concentration from 14 to 21 days is associated with an increase in the total number of microorganisms. Therefore, the rates of their destruction increased. The bacterial community actively participated in bio-drying, and the change in external conditions affected its diversity, abundance and succession. Apparently, the role of archaea did not affect the studying material. This observation suggests that during bio-drying, the conditions for these organisms were unsuitable.

From the beginning of the bio-drying process, the *Ascomycota* dominated (94% of all micromycetes). That was also noticed during the composting of AnWS [12]. *Ascomycota* members are widespread in most terrestrial ecosystems (such as genera *Aspergillus*, *Penicillium*, *Saccharomyces*, etc.) and play a key role in the decomposition of organic residues. Jointly with other divisions of fungi, they are involved in lignin degradation [43]. However, *Ascomycota* were the dominant division of the fungal community during the entire process of composting WS [17] and food waste [13]. The activity of *Basidiomycota* members appears only on day 14 (5%), which is due to a lack of necessary nutrients and contention with other fungi. Members of the *Basidiomycota* are of great ecological importance due to their

ability to process complex biopolymers, especially lignin [3]. *Pseudogymnoascus* members (*Ascomycota*) were found at all stages of bio-drying. They accounted for 11.6% at the beginning of the process. Then, their presence dramatically increased to 99.5% on day 7, while communities changed. Furthermore, on day 14, it dropped to <1%. The dramatic increases and decreases in the number of *Pseudogymnoascus* are due to the fact that some of its members are adapted to alkaline conditions where they can oxidase aromatic hydrocarbons [44]. Probably, these *Pseudogymnoascus* members in the current study are heat-resistant, since their vital activity did not stop, and they dominated at high temperatures (50–52 °C) on days 7, 21. During screening of lignolytics, Hossein et al. [45] found out that this genus had the highest cellulolytic and ligninolytic activity among all fungal isolates. The presence of *Pseudogymnoascus* species in the substrate was due to their ability to destroy lignocellulose of the wood chips. However, the lowest number of ITS copies μg$^{-1}$ was on day 7. Thus, not one fungi had yet adapted to such conditions. Fungi of genus *Mucor* (*Mucoromycota*) accounted for 48.9% of the total fungal community on day 14 and 12.8% on day 21. Moreover, their greatest number was on day 14, when the number of *Pseudogymnoascus* species was minimal, presumably due to mutual antagonism. Thermotolerant *Mucor* species are characterized by the ability to decompose xenobiotics at a temperature of 58 °C [46].

The highest amount of the genus *Penicillium* was found in the initial mixture (8.1%) and on day 21 (3.0%). In a study by Bian et al. [47] *Penicillium* members had similar dynamics during the bio-drying of WS that was formed in chemical fiber production plants. The microorganisms of genera *Humicola* 50.1%, *Enterocarpus* 15.2% and *Mortierella* (*Mucoromycota*) 5.8% probably got into the initial substrate with wood chips, which served as their habitat due to the ability of some members to break down hemicelluloces. However, they could not adapt later to the increase in temperature and pH of the environment and reduced their quantity to <1%. It was shown that members of the genus *Mortierella* were dominant at the cooling stage [46], and *Enterocarpus* accounted for nearly 1% at a temperature of 45 °C during composting of anaerobically digested municipal solid waste [48]. On day 14, the abundance of the genera *Melanocarpus* and *Guehomyces* (*Basidiomycota*) was maximum and accounted for 5.4 and 4.7%, respectively. During this period, the temperature varied within 50.6–52.2 °C. Perhaps, the decrease in moisture content during this period led to the inhibition of their growth. Similar results have been previously shown by Feng et al. [49]. Thus, fungi of the genus *Melanocarpus* (less than 5%) were found during composting of rice straw at neutral acidity in the maturation stage. Wang et al. [17] noticed that during the composting of WS, genus *Guehomyces* were encountered throughout the entire process in an amount from 1 to 5%. Yeasts of genus *Debaryomyces* (37.2%) have been described as dominant in the thermophilic stage during the composting of olive oil waste [50]. Members of the genus *Orbicula* on day 21 accounted for 25.9% of the fungal community, while it had been <1% at the previous stages on days 7 and 14. Thus, the *Orbicula* fungus is quite resistant to elevated temperature (52 °C) and lack of moisture. Members of this genus have been found on substrates such as wood, leaves and straw [51]. Members of the genus *Gymnascella* showed an active growth (4.4%) only on day 21. In the original substrate, their amount was 0.02%. Many species of *Gymnascella* are an important link in the decomposition of hardly decomposable substrates, including keratin [52]. Therefore, an unsuitable substrate could limit the dynamics of their development.

The diversity of bacteria decreased after the onset of thermophilic stage on days 7–14. Thus the temperature conditions determined the possibility of survival and use of the substrate and, hence, the composition of the bacterial community [5]. Members of the *Bacillota* phylum were the major part of the bacterial community during the most intense period (7–21 days). The abundance of the *Pseudomonadota* community was the highest at the beginning of the thermophilic stage. In the initial mixture, the second largest (25%) were members of the *β-Proteobacteria* class. Then, their amount dropped to 3% on day 7 and to 1% on day 14. At the same time, it is reported that these microorganisms can live in many terrestrial and marine environments, including extreme conditions such as hydrothermal vents [15]. Members of the *Actinomycetota* phylum were found during bio-drying, and their

amount varied from 2 to 6%. This indicates their ability to maintain inside substrate and participate in the process, but to a small extent. According to a study by Steger et al. [53], fungi and actinobacteria usually proliferated during periods of relatively low temperatures (<45 °C). Similar data on the dominance of the genus *Prevotella* during composting a mixture of soil with fruit waste have also been reported [54]. Changes in temperature conditions and pH led to decrease in the initial amount of *Prevotella* and *Simplicispira* members. Piceno et al. [55] noted that in the thermophilic stage of WS composting, these genera were not detected, although they were found in the initial mixture. It was reported that members of *Prevotella*, which prevail in the soil, take an active part in the degradation process and get into the compost mixture through the materials used [54]. The genus *Shuttleworthia*, which was in the original biomass, usually inhabits the activated sludge. Recent studies have shown that members of this genus are dwellers in cattle rumen, as well as starch and sugar degraders [56,57]. Therefore, an increase in temperature above 40 °C is critical for these microorganisms. They are suitable exclusively for mesophilic conditions and a certain nutrient substrate, which can explain the inactivation of the *Shuttleworthia* species on day 7. Members of the genus *Pseudomonas* were also found only in the original substrate. In a study on the composting of pulp and paper waste, Beauchamp et al. [58] noticed that an activity of the *Pseudomonas* species was at temperatures from 18 to 25 °C, and the activity also depended on the carbohydrate source used. Thus, the activity of this bacterial group terminated due to temperature rise above 50 °C on day 7 and alkalization of the medium. The initial mixture also contained few members (each less than 2%) of such genera as *Stenotrophomonas*, *Eoetvoesia*, *Lysobacter*, *Neochlamydia* and *Enterococcus*, which were not further observed. Al-Dhabi et al. [59] noted that mesophilic *Stenotrophomonas* species are capable of denitrification and are involved in the treatment of industrial wastewater. Members of the genus *Eoetvoesia* are aerobic mesophilic bacteria not capable of denitrification and prefer a slightly acidic or neutral environment (pH 6–7). *Eoetvoesia* species have been isolated from the wastewater treatment system [60]. In a study on composting of WS and other materials by Vaz-Moreira et al. [61], members of *Lysobacter* were also found rarely. *Neochlamydia* members were found mainly in the activated sludge layer but did not dominate [62]. *Enterococcus* species were in WS and in untreated wastewater [63]. Thus, these species typical for WS microorganisms disappeared during bio-drying, probably due to unsuitable conditions for them on days 7–14: slightly alkaline pH and a dramatic increase in temperature.

Diversity of the genera taxa consistently decreased during bio-drying. In the thermophilic stage in the period from 7 to 14 days, the dominant bacterial genera were *Bacillus* and *Amaricoccus*, the proportion of which increased from 43.5 to 84.6% and decreased from 15.7 to 2.2%, respectively. A significant increase in the number of *Bacillus* was probably facilitated by an oxygen-limited environment. During the bio-drying sections *f*–*g* and *i*–*j* in Figure 2, biodegradation occurred intensively at a temperature of 52–54 °C, while *Bacillus* have been dominating. So, in a study by Cai et al. [5], *Bacillus* spp. dominated in the thermophilic stage on days 11–15 at a temperature of 50.3 °C. *Bacillus* spores are also extremely resistant to moisture coupled with heat and typically require 80–110 °C to destruct them rapidly [64]. It has been reported that changing conditions inside treatment tanks from anoxic to aerobic resulted a change in the metabolism of *Bacillus* members [65]. In general, over the entire period of bio-drying, the oxygen concentration in the waste air averaged near 17.5 vol.% with a minimum value of 12.8 vol.%. The presence of $H_2S$ in the waste air indicates about the creation of microaerophilic or even anaerobic conditions during the entire bio-drying process. During the periods of detection of the minimum $O_2$ concentration in the waste air, $O_2$ was completely absent in the dense particles of the substrate. Probably for this reason, aerobic processes occurred simultaneously with fermentation and anaerobic respiration with the formation of $H_2S$. Although genetic analysis showed the presence of *Desulfovibrio* and *Desulfosporosinus* sulfate reducers, their total number dropped from 1.0% to less than 0.1%. However, the final reaction with the formation of $H_2S$ can also be done by sulfur reducers belonging to the genera *Coprothermobacter* and *Clostridium*, the number

of which exceeds 2.5–1.3% on days 7–21. Members of the genus *Amaricoccus* were at their maximum abundance on day 7 (15.7%), when active formation of nitrate took place (the maximum was 3134 mg kg$^{-1}$). It was, probably, due to the oxidation of nitrogen-containing compounds by nitrifying aerobic bacteria. Subsequently, on day 14, their amount decreased to 2.2%. The nitrate concentration decreased along with the decrease in the abundance of *Amaricoccus*. Some *Amaricoccus* species belong to nitrite-oxidizing members of the *Alphaproteobacteria* class [66]. Maszenan et al. [67] noted that members of the genus *Amaricoccus* were isolated from wastewater, where they grew at the temperatures from 20 to 37 °C and pH values from 5.5 to 9.0.

It is possible that such temperature dynamics and low moisture content led to the creation of specific conditions in the period from 7 to 21 days and the development of some microorganisms, the proportion of which did not exceed 3% and remained practically unchanged. Members of the genera *Clostridium* and *Romboutsia* on day 14 accounted for 1.3 and 1.7%, respectively. *Clostridium* spp. are strictly anaerobic bacteria whose most common habitat is manure and compost in the cooling and maturation stages. Thus, their activity was inhibited in the thermophilic stage [68,69]. *Romboutsia* and *Clostridium* members were identified in manure, but their proportion decreased with composting [70]. On day 21, the bacterial community changed with the appearance of new genera: *Saccharopolyspora* 3.7% and *Halopolyspora* 1.1%. Zhan et al. [71] detected that the amount of genera *Halopolyspora* and *Saccharopolyspora* members was near 1% during the thermophilic stage. *Saccharopolyspora* members have been found in the composting of WS and plant wastes, and their ability to produce antibiotic compounds has been noted, which allows them to compete with other bacteria [72].

When analyzing the taxonomic diversity of the microbiota of a biofilter, it is necessary to take into account the possibility of constant entry of microorganisms into the biofilter from the reaction chamber with the waste air flow. The dominance of members of genera *Sphingobacterium* (*S. multivorum*), *Pseudomonas* (*P. putida*, *P. fluorescence*, *P. testosteroni*), *Achromobacter*, *Brevundimonas*, *Rhodococcus*, *Acinetobacter* has been reported in biofilters during the decomposition of aromatic compounds: benzene, toluene, ethylbenzene and p-xylene [73]. Some species of *Sphingobacterium* sp. and *Pseudomonas* sp. are responsible for the removal of hydrogen sulfide, sulfur dioxide and mercaptans formed during the decomposition of WS [74]. Bacteria of these genera could remove H$_2$S formed during the bio-drying of WS. Li et al. [75] also found members of the genus *Brucella* in biofilters under similar conditions and in small amounts, but their role in the biodegradation of VOCs has not yet been elucidated. However, Liu et al. [76] showed that *Brucella* species are active participants in treatment of inorganic sulfur compounds in wastewater. There are aerobic denitrifies among the genera *Brevundimonas*, *Achromobacter*, *Pseudomonas* and *Bacillus* [77–79], and most of the identified genera in the biofilter can remove ammonia via assimilation. *Fusarium* fungi are practically the only ones in the biofilter (99.6%). It is reported that *Fusarium solani* were inoculated into biofilters and were able to oxidize toluene [80], methane [81], pentene, hexane [82] and dimethyl sulfide [83]. The presence of the above-mentioned microorganisms in biofilter provided removal of VOCs from the waste air from WS bio-drying by more than 90%.

In just 21 days of bio-drying, the substrate moisture decreased by 19.7%. Some similar dynamics of WS bio-drying (11.4% in 20 days) are reported by Cai et al. [5]. The temperature dynamics during the bio-drying of WS differed significantly from the temperature during composting of AnWS [12]. Instead of the dependence of temperature on time characteristic of composting, as in the case of AnWS, there was no pronounced maximum with a further decrease in the bio-drying of WS. The duration of the high temperature period (50–54 °C) increased from 2–3 days to 2 weeks, which could be due to the higher content of readily decomposable organic matter available in WS compared to AnWS. WS inhibited the growth and development of the test plant [33]. This is due to the increased content of heavy metals. For example, the Zn and Cr$^{3+}$ content exceeds the permissible hygienic standards established for clean soil.

## 5. Conclusions

Thus, the results of this research showed that the *Bacillus* members were the main decomposers of WS organic substances. The period from 7 to 14 days can be characterized as the most effective, when the OM mineralization and moisture loss were the highest. During the transition from the mesophilic stage to the thermophilic one, the decomposition of organic substances slowed down. The periods of this significant decrease in the rate of biodegradation were noticed, which is probably due to succession of the initial microbial community of WS and is a reserve for increasing efficiency. The presence of a community of microorganisms in the biofilter ensured the removal of VOCs from the waste air of the bio-drying WS by more than 90%. Taken together, the obtained results provide a theoretical basis for developing strategies to improve the efficiency and safety of WS bio-drying.

**Author Contributions:** Conceptualization, V.M.; methodology, V.M., N.Z. and V.Z.; software, I.M.; validation, V.M. and I.M.; formal analysis, V.M., A.S. and I.M.; investigation, V.M., A.S. and I.M.; resources, V.M. and N.Z.; data curation, V.M. and V.Z.; writing—original draft preparation, V.M., A.S. and I.M.; writing—review and editing, V.M., A.S. and I.M.; visualization, V.M., A.S. and I.M.; supervision, V.M.; project administration, V.M.; funding acquisition, V.M. All authors have read and agreed to the published version of the manuscript.

**Funding:** This research received no external funding.

**Institutional Review Board Statement:** Not applicable.

**Data Availability Statement:** No new data were created or analyzed in this study. Data sharing is not applicable to this article.

**Conflicts of Interest:** The authors declare no conflict of interest. The funders had no role in the design of the study; in the collection, analyses, or interpretation of data; in the writing of the manuscript, or in the decision to publish the results.

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
