# Peer review of "Bio-Drying of Municipal Wastewater Sludge: Effects of High Temperature, Low Moisture Content and Volatile Compounds on the Microbial Community"

_fermentation, doi:10.3390/fermentation9060570_

Round 1

Reviewer 1 Report

The manuscript reports on the impact of drying process on sludge microbial community and the physical and biochemical properties. Some comments that need to be addressed are as below:

1)In the title, do authors mean moisture content or humidity?

2)What was the humidity of air (in the room) flowing through the chamber?

3)The drying setup needs to be presented in this manuscript.

4)Why no (statistical) experimental design was used in this study?

5)L90: Please include the number for desired porosity in the text.

6)L127: Why such a high limit of error (10%) was considered as the measurement error in GC analysis?

7)L198: ‘chemical reactions’ should be replaced by ‘biochemical reactions’.

8)The ‘toxicity’ terminology should be explained/clarified in the text.

9)Error bars in all graphs need to be described what they are (in the title of each graph)? Are they standard error or standard deviation?

English is fluent and statements are easy to understand.

Reviewer 2 Report

Biodrying is a kind of waste pretreatment technology developed from composting technology. Biodrying of sludge moisture removal has engineering application value. But this article needs some work.

1)       The title doesn't quite fit the content of the article

2)       The use of dried sludge is not clearly explained in the introduction.

3)       Germination rate doesn't mean much here.

4)       A schematic diagram of Experimental setup is suggested

5)       The raw material parameters listed in Table 1 are in the optimal range. It is suggested to list the corresponding residue parameters after 21 days.

6)       According to Figure 1, why did the author choose only a 21-day experimental period? Which coordinate does the water content in Figure 1 correspond to?

7)       According to the experimental results in Figure 2, H2S and NH4 have great changes. The discussion section has not discussed these changes and the changes of microorganisms.

8)       Figure 4 shows the change of fungle communite over time. Please explain this phenomenon in the discussion section.

Round 2

Reviewer 2 Report

Bio-drying of municipal wastewater sludge: effects of high temperature, low moisture content and volatile compounds on the microbial community

This paper is about the change process of microbial community in the biological drying process of municipal wastewater sludge, which has good research significance for the pretreatment of organic waste, but there are still some changes in the paper.

3 Results

At line 302, the dash after bacteria needs to be removed

Part 3.1.2,the description of changes in microbial community biomass needs to be revised, and the current description is confused.

Part 3.2.2, the description of fungal community changes in this part is not detailed enough. In addition, at what stage "the dominant genera of the fungal association were Pseudogymnoascus and Mucor" is not described. As can not be seen from Figure 5, these two genera of fungi belong to the dominant species in the whole process.

Conclusion: Accept after minor revision

Moderate editing of English language required
